# Nutritional Evaluation of Black Soldier Fly Frass as an Ingredient in Florida Pompano (*Trachinotus carolinus* L.) Diets

**DOI:** 10.3390/ani12182407

**Published:** 2022-09-14

**Authors:** Amiti Banavar, Samad Keramat Amirkolaei, Lexi Duscher, Bela Haifa Khairunisa, Biswarup Mukhopadhyay, Michael Schwarz, Steve Urick, Reza Ovissipour

**Affiliations:** 1Future Foods Lab and Cellular Agriculture Initiative, Virginia Seafood Agricultural Research and Extension Center, Virginia Polytechnic Institute and State University, Hampton, VA 23699, USA; 2Department of Food Science and Technology, Virginia Polytechnic Institute and State University, Blacksburg, VA 24061, USA; 3Department of Biochemistry, Virginia Polytechnic Institute and State University, Blacksburg, VA 24060, USA; 4Virginia Seafood Agricultural Research and Extension Center, Virginia Polytechnic Institute and State University, Hampton, VA 23699, USA; 5Department of Biological Systems Engineering, Virginia Polytechnic Institute and State University, Blacksburg, VA 24060, USA

**Keywords:** alternative ingredients, fish, frass, growth, gut microbial community, insect

## Abstract

**Simple Summary:**

Carnivorous fish species, such as the Florida pompano, currently require expensive and unsustainable feed in order to be raised in an aquaculture setting. A waste product of the black soldier fly, frass has the potential to be an alternative carbohydrate source for these fish. This study aims to assess the potential of frass to mimic the growth, body, composition, and intestinal microbiome of pompano that can be achieved through commercial feed. Three diets with varying frass levels as well as a control diet were developed. The frass was found to increase the visceral somatic index and lower the hepatosomatic index. The frass did not alter the body composition or improve the growth performance, leading to a lower specific growth rate and higher feed conversion rate. The microbiome analysis showed the highest diversity of the gut flora in the control diet, while the frass diets showed signs of community imbalance. This may have been due to the decreased starch in the frass, which is usually metabolized by those microbial communities. Overall, frass diets are not ideal for carnivorous fish diets, but have could have potential for feed replacement in herbivore and detritivore fish species.

**Abstract:**

The aquaculture industry is in need of sustainable fish feed to reduce the use of expensive and environmentally invasive wild-caught fish currently fed to many carnivorous species. The black soldier fly (BSF) has become a popular sustainable alternative protein source; however, the nutritional waste byproduct of BSF, frass, has not been extensively studied as a feed replacement in carnivorous species. This study evaluates the potential of BSF frass on the growth, body composition, and intestinal microbiome of the Florida pompano, *Trachinotus carolinus*. Four experimental diets were formulated containing different levels of frass, replacing plant-based carbohydrate sources. As a result of this study, the frass did not improve the growth performance, resulting in a lower specific growth rate and higher feed conversion rate. While the frass diets did not alter the body composition, the visceral somatic index (VSI) significantly increased compared to the control diet and the hepatosomatic index (HIS) was lowered. The microbiome analysis showed high variation among the diets, with the control diet having the most distinct consortia, which may have been driven by the increased levels of starch compared to frass diets. This study indicates that BSF frass may not be a suitable feed replacement for carnivorous pompano; however, frass could still potentially be a replacement feed for herbivore or detritivore fish and should be further studied.

## 1. Introduction

The aquaculture industry is rapidly growing and is dependent on the supply of efficient and sustainable aquaculture feeds [1,2]. The partial or complete substitution of expensive protein sources in current aquaculture feeds with cheap byproduct feed ingredients would decrease the diet costs, as well as alleviate the environmental impact of wild-caught feeds. Moreover, the increasing competition for feed ingredients such as soy, wheat, corn, and barley between the aquaculture, poultry, and cattle industries, as well as for biofuel, may threaten the sustainability of aquafeed production [3]. Animal and plant byproducts that are not yet fully exploited have great potential to meet these limitations and enhance sustainable and low-cost feed production.

An animal byproduct that has great potential in feed replacement is frass, a waste product obtained by growing black soldier fly (BSF) larvae on various substrates. Recently, insect larvae meal has received significant attention as a novel alternative protein source in aquafeed [4,5,6,7]. However, less work has been done with insect byproducts, such as frass, as feed replacements. The frass byproduct from BSF is normally made up of the waste matter from insect feed, dead larvae, and larvae excretion, which altogether contains around 200 g/kg protein and significant contents of essential nutrients [8,9]. Previous studies have suggested that the addition of frass has a positive impact on vegetable growth, and that it can be applied as a functional biofertilizer [10,11]. Frass also contains chitin and beneficial microbes, which may have a positive influence on the intestinal tract health of certain fish [8,12]. Recent published data on the addition of frass in catfish and tilapia diets showed that dietary frass can improve the performance and immune systems of fish [8,9,13].

The Florida pompano is a popular marine aquaculture fish mainly due to its excellent taste, rapid growth with commercial diets, relatively high sales price, and ability to adapt to seawater with varying salinity levels [14,15,16]. Pompano farming has been launched in the United States because of the significant consumer demand. Nevertheless, pompano production is still limited due to the lack of available information about its nutrient requirements [17,18]. A well-balanced diet with reasonable prices is necessary to guarantee the growth of the pompano aquaculture industry. 

Although the larval frass from BSF byproduct had a significant positive effect on the performance of channel catfish [8] and hybrid tilapia [9], little to no research has yet been conducted to investigate the effect of frass as an ingredient in carnivorous fish, including pompano. Moreover, frass beneficial microbes [12,19,20] may be supplements to antibiotics or probiotics used to control pathogens and ensure animal health in the aquaculture industry. Thus, this study focuses on the potential of frass as an emerging ingredient in pompano feed by assessing the effects of dietary frass at different levels on growth parameters, the whole-body composition, and the intestinal microbiome of pompano.

## 2. Materials and Methods

### 2.1. Ethics Statement

This experiment was approved by the Animal Care and Use Committee of Virginia Tech (Protocol number IACUC # 20-113).

### 2.2. Feed Preparation

Lindo software was used to formulate diets, including a control diet containing 43.5% crude protein and 9% crude fat using common ingredients for aquafeed (Table 1). Three experimental diets with different levels of frass (Table 2) (6, 12, 18%) were formulated by replacing a mixture made up of equal parts of corn, wheat, and soybean meal. The experimental diets were prepared to meet the essential nutrient requirements of pompano. All ingredients were finely ground, mixed, and pelletized. A pellet maker with a 3-mm-diameter die was used to produce the diets. All four diets were dried at 45 °C at the same time and stored at −20 °C until use.

### 2.3. Experimental System and Animal

This study was carried out at the experimental facility of Virginia Seafood Agricultural Research and Extension Center (AREC) in Hampton, VA. Ten juvenile pompano individuals with an average initial body weight of 31.14 ± 0.32 g were randomly distributed into each of the 12 (300 L) tanks on the same recirculation system equipped with fluidized-bed biofilters, bubble bead filters, protein skimmers, UV sterilization, immersion titanium heaters, and a diffusion aeration system. The fish were adapted to a commercial diet and their tank environments for a week before the start of the experiment. After adaptation, the whole fish were weighed and returned to their same experimental tank.

There were four diets randomly assigned to three tanks, each in a completely randomized tank design. The fish were hand-fed at a restricted feeding level (3.4% of body weight/day) twice a day (8 AM and 5 PM) for 8 weeks. The restricted level selection was based on the feed intake level of pompano fed to an apparent similar weight range [21,22]. The fish biomass was measured every two weeks to adjust the feeding level. The photoperiod was held at 12 h light:12 h dark with fluorescent lights controlled by electric timers. The water quality parameters such as the water temperature, salinity and dissolved oxygen, and pH were monitored daily throughout the study. The total ammonia, nitrite, and nitrate–nitrogen levels were quantified twice a week via spectrophotometric analysis.

### 2.4. Sampling Procedure

At the end of the eight weeks of feeding, all fish were weighed to the calculate final weight, specific growth rate (SGR), and feed conversion ratio (FCR). Three fish were randomly selected from each tank and euthanized according to the previously established IACUC (Institutional Animal Care and Use Committee) procedures using overdosed tricainmethane sulfonate (Sigma-Aldrich, St. Louis, MO, USA) (MS-222, 100 mg/L, buffered to pH 7.4) for the whole-body composition analysis. Another three fish were used for the calculation of visceral somatic and hepatosomatic indices by separately weighing whole viscera and the viscera separated from the liver. Prior to dissection, the fish were rinsed with sterilized distilled water, cleaned with ethanol (70.0%), and washed again with sterilized distilled water to eliminate all exterior bacteria. In addition, whole intestinal tracts were sterilely removed from these fish and the gut digesta and mucosa were collected from sterile scrapings from the mid-hindgut section. There were three fish collected from each of the three treatment replication tanks, resulting in nine fish gut samples collected for each feed treatment. All euthanasia and gut content collection procedures were completed in less than three hours.

### 2.5. Whole-Body Sample Analysis

The experimental feeds and whole-body fish samples were analyzed for their proximate composition. The samples were dried in a convection oven at 105 °C for 12 h to determine the moisture level. The dried samples were finely ground with a mortar and pestle and thoroughly homogenized to obtain representative sub-samples. The samples were analyzed for crude protein (total nitrogen × 6.25) using the combustion method with a nitrogen determinator (Elementar nitrogen analyzer, Ronkonkoma, NY, USA). The crude lipids were analyzed by extracting them with petroleum ether as the extracting solvent in an ANKOM XT15 extractor (ANKOM Technology, Macedon, NY, USA). The ash content was analyzed via incineration at 550 °C in a muffle furnace for 5 h. The starch content of the diet was measured enzymatically according to the method described by Goelema et al. [23]. The mineral measurement was performed using the inductively coupled plasma mass spectrometry (ICP) method based on the standard NEN 15510. The amino acids of the frass were determined by hydrolyzing the samples for 16 h at 130 °C in HCl (vapor phase), following by derivatization with Waters AccQTag derivatization reagents (Milford, MA, USA). The quantification process was conducted using RP UPLC, with a C18 analytical column (1.7 µm, 2.1 × 100 mm) and acetonitrile and water as buffers.

### 2.6. DNA Extraction and 16S Analysis

The DNA was extracted from each gut content replicate using a QIAamp Fast DNA Stool Mini Kit (Qiagen) with additional bead beating for the lysis of the bacterial cell walls. The DNA was treated with RNAse to remove the contaminating RNA. The DNA quantity was measured using a Qubit Fluorometer and the quality was determined via gel testing. The 16S V4-V5 region of each DNA replicate was barcoded and PCR-amplified with forward primer 515 and non-barcoded reverse primer 926 in triplicate to account for the amplification bias [23]. All PCR products were purified and pooled in equal concentrations before sequencing on Illumina MiSeq V2 600-Cycle Cluster at the Virginia Tech Genomics Sequencing Center.

Demultiplexed and trimmed paired-end reads were obtained from the Virginia Tech Genomics Sequencing Center. These sequences were analyzed using a modified eASV pipeline [24,25] and QIIME 2-2019.4 [26]. Briefly, bbsplit.sh [24,25] was used to separate the 18S and 16S rRNA sequences by mapping the amplicons to SILVA 132 [27] and PR2 [28] databases. Each of the pooled reads were imported to QIIME2 to generate Amplicon Sequence Variants (ASVs) via the DADA2 pipeline [29] with clustering at 99% sequence similarity using vsearch [30]. A pre-trained Naïve–Bayes classifier specific to the 515F and 926R primer sets [31] was used to annotate the ASVs using the SILVA 138 database [27]. ASVs annotated as chloroplasts, mitochondria, and Cyanobacteria are categorized as contaminants and were removed from the dataset [32,33,34].

The ASV dataset was exported from QIIME to R [35] for further diversity and phylogenetic analyses using Bioconductor packages [36]. Prior to the analysis, all technical replicates were pooled, resulting in three biological samples from four treatment groups. Singletons and doubletons, as well as ASVs that were unassigned at the phylum and class levels, were removed from the dataset. The alpha diversity estimator Shannon (for the species diversity and evenness of the microbiome, as well as estimates of the effective number of species), Chao1 (species richness), and Simpson (proportion of species in a sample, dominance index in the community) were calculated using the microbiomeSeq package [37] with samples rarefied to 712 sequences per sample. A pairwise ANOVA (*P*_ANOVA_ < 0.05) was used to determine the significance of the differences between the alpha diversity metrics for each treatment. A microbial community comparison between samples was performed via principal coordinate analysis (PCoA) ordination of the unweighted UniFrac [38] distances using relative abundance-transformed counts. A phylogenetic analysis at the phylum level was conducted and twenty of the most abundant genera were also analyzed via phyloseq [39] with averaging of the ASVs within the feeding groups.

### 2.7. Statistical Analysis

The weight gain was determined by the difference between the initial and final body weights. The feed conversion ratio (FCR) was calculated per tank from the feed intake and weight gain data:FCR = feed consumed (g)/wet body weight gain (g)

The protein efficiency ratio (PER) was calculated per tank from the weight gain and crude protein feed data:PER = Wet weight gain (g)/crude protein consumed (g)

Specific growth rate (SGR) was calculated as follows and expressed as a percentage:SGR = 100 (Ln W1 − Ln W0) × days^−^^1^
where W1 was the final weight and W0 was the initial weight. 

The hepatosomatic index (HSI%) and visceral somatic index (VSI%) were calculated as below:HSI% = 100 × (liver weight/whole bodyweight); VSI% = 100 × (Visceral weight/whole body weight)

The data are presented as the means of each treatment with a standard deviation. The percentage data were arcsine-transformed to achieve normality and the normal distribution of the data was verified using the Kolmogorov–Smirnov test. All data were analyzed using a one-way ANOVA for the effect of the dietary frass on the growth parameters and whole-body composition. Tukey’s multiple comparison test was used to determine the statistical differences among the treatment means.

## 3. Results and Discussion

### 3.1. Growth Performance

The 8-week growth trial demonstrated that the growth performance parameters of pompano were affected by the inclusion of the frass (Table 3; *p* < 0.05). The replacement of corn and wheat flour with the frass resulted in a lower final weight (*p* < 0.05). Maximum weight gain was observed in fish fed a control diet with no frass. The current data reveal that dietary frass does not appear to be a beneficial ingredient for pompano growth and that the growth performance is highly dependent on the type and consequently the nutrient composition of the frass. The frass used in the current study had lower contents of protein (106 vs. 216 g/kg) and fat (8 vs. 63 g/kg) compared with the frass obtained from BSF larvae grown on distillers’ dried grains as a substrate [8]. In this study, we used frass from insects that were fed on hemp waste with significantly lower protein contents. These nutrients are vital for metabolism and growth and may be a main factor in the decreased growth of fish fed graded levels of frass.

The biweekly weight measurements revealed that there were no consistent differences among the diets in first four weeks, but the growth began to differ distinctively in favor of the control diet from week 6 of the experiment as seen in (Figure 1). In contrast to our results, two recently published papers indicated that the growth parameters improved significantly in channel catfish [8] and in hybrid tilapia [9] fed diets containing frass at levels from 100 to 300 g/kg. These inconsistent results may be caused by differences in the feeding habits and size of the gastro-intestinal tract between omnivorous species (catfish and hybrid tilapia) and carnivorous species (pompano). The high fiber content in frass is not digestible for carnivorous fish such as pompano that have short intestines, meaning it has little nutritional value for these species. Previous work that evaluated the effects of dietary fiber on fish growth suggested that the supplementation should be restricted to less than 7% for European sea bass [40] and 8.5% for largemouth bass [41]. Moreover, the National Research Council [42] suggested that the dietary fiber level of the fish diet should not exceed 10%. The pompano appears to be more sensitive to increased fiber contents, despite the relatively low dietary fiber content of the frass used in this study (2.7 to 5.1%), which may have contributed to the negative impact on the pompano individuals’ growth compared to the control diet.

Along with the growth, the FCR and SGR were also significantly lower in pompano fed frass diets in comparison to the control diet (Table 4; *p* < 0.05), and these decreases were correlated with increasing levels of frass. The protein efficiency ratio (PER) was also lower in fish fed frass diets than those fed the control diet (Table 4; *p* < 0.05). An impaired FCR and a lower PER with increasing frass levels suggest that high dietary fiber may result in a low nutrient utilization efficacy in pompano. These results are similar to other studies that observed growth depression mainly because of the low nutrient digestibility of cellulose diets [43,44,45]. Similarly, Maas et al. [46] found that digestibility reduces by up to 4.4% with each 1% addition of dietary fiber. In addition to low digestion, indigestible fiber can increase the intestinal evacuation rate and lower the retention time of digestible nutrients in the gut, thereby lowering the nutrient absorption capability in carnivorous species [47].

The lower feed utilization rate with increasing frass levels may also be related to the larger content of poorly digestible elements, such as ash and chitin. The dietary ash content increased from 75.8 g/kg in the control diet to 102.9 g/kg in the 18% frass diet. The impacts of poorly digestible feed components on the digesta movement may modulate nutrient digestion and absorption. Similar to fiber, the high ash content of frass diets can reduce the digesta retention time in the gastro-intestinal tract, thereby limiting the association of enzymes with nutrients [8]. This condition may decrease the nutrient digestibility in high-frass diets. Similarly, the feed efficacy ratio in catfish was reduced with the increasing dietary ash caused by the frass addition [8]. Therefore, the abundance of poorly digestible fiber and ash along with the fast evacuation rate of the digesta may explain the lower feed efficiency in pompano fed frass diets.

In addition to the quality of the frass, the feeding level has a significant impact on the fish growth. In the previous studies with frass, fish were fed at satiation feeding level, and the growth enhancement was correlated with an increase in feed intake in catfish and hybrid tilapia [8,9]. Fish can adjust themselves to increased dietary fiber by enhancing their dietary consumption to fulfill their energy requirements. This ability was previously reported for rainbow trout [48] and channel catfish [49]. These results suggest that the lower nutrient availability in frass diets was compensated for by a greater feed intake and resulted in similar growth in catfish and hybrid tilapia [9]. Therefore, the greater growth with high-frass inclusion treatment (30%) in a previous study [8] may have mainly been a result of larger feed consumption.

The VSI and his were also affected by the dietary frass inclusion. The VSI was larger in pompano fed on the diets containing 12 and 18% frass in comparison with the other diets (Table 4; *p* < 0.05). The dietary frass reduced the liver weight in the pompano, and the control diet without frass led to a larger HSI value (Table 4; *p* < 0.05) in comparison with fish fed frass-containing diets. A lower HIS indicates reduced liver weight and energy stores in the fish; considering there is less availability of carbohydrates in frass to store as fat, this is consistent with previous work on the effect of the carbohydrate content on the HSI in carnivorous fish [50,51].

### 3.2. Whole-Body Composition

The analysis of the body composition showed that the dietary frass did not change the proximate body composition of the pompano fish. The moisture, protein, and fat values ranged from 67.9, 18.3, and 9.0 to 68.9, 19.0, and 9.9%, respectively, and were statistically similar for all four treatments (Table 5; *p* < 0.05). These results were similar to those found by Yildirim-Aksoy et al. [8,9], who observed no body composition differences in channel catfish and hybrid tilapia fed by BSF frass at significantly higher inclusion rates than in our study. The large fiber and low starch contents of the dietary frass were thought to limit the nutrient and energy availability in pompano and to change their body composition. However, the body fat content in pompano is not affected by the availability of dietary energy. Thus, it seems that the lower nutrient and energy availability in frass treatments resulted in a lower growth rate in pompano rather than affecting the proximate composition. Despite the higher mineral contents of the frass diets, the whole-body mineral contents were similar for all four diets (Table 3 and Table 5; *p* < 0.05). The lack of mineral composition differences may have been caused by the poor utilization or digestion of frass by the pompano fish.

### 3.3. Microbial Composition Analysis

The sequencing of 16S rRNA V4-V5 amplicons generated a total of 172,355 sequences, which corresponded to 857 ASVs. The set with a dietary inclusion of frass at the 12% (*w*/*w*) level showed the lowest species richness (i.e., Chao1) among all treatment groups, while the control group had the greatest number of ASVs (Figure 2). All groups shared similar levels of species evenness (i.e., Shannon), except the feed with 6% frass, in which some species dominated the consortia, as shown by the low Shannon diversity and high Simpson dominance index (Figure 2). However, a pairwise ANOVA showed that these differences were not significant, indicating that the frass addition did not alter the species richness of the pompano gut microbiome (Figure 2; *p* > 0.05).

The frass inclusion altered the composition of the pompano gut microflora (Figure 3a–c). The samples belonging to frass dietary groups displayed separate clustering from the control group, as shown in the PCA analysis (Figure 3a). This distinction was clear even at the phylum level (Figure 3b) and was further clarified at the genus level (Figure 3c). However, there was still a high level of variability among the replicates and any separation was explained only by the very low principal components (<13%; Figure 3a). Regardless of the feed, Firmicutes, Proteobacteria, Bacteroidetes, and Actinobacteria contributed up to 93% of each of the pompano gut microbiomes. The composition was similar to previously reported fish microbiome studies [52,53,54]. In general, the dietary frass lowered the abundance level of *Bacteroides* species (Figure 3b,c), which were most enriched in the control group, as well as those of the *Lactobacillus, Veilonella,* and *Bifidobacterium*. The observed decrease in *Bacteroides* and *Bifidobacterium* abundance levels may have been directly related to the reduced starch percentage in the diets due to the frass supplementation (Table 1), as many species within these genera actively degrade starch and other available polysaccharides in animal gastrointestinal tracts [55]. *Bacteroides* species metabolize starch via a starch utilization system (Sus) [56,57], while *Bifidobacterium* species perform this process via extracellular glycosyl hydrolases (GHs) [58]. *Lactobacillus* and *Veillonella* contain known probiotic species that are commonly found in fish gut [59,60]. The frass addition enriched *Vibrio* and *Listeria* species of the Proteobacteria and Firmicutes phyla, respectively. The increase in *Vibrio,* one of the most common genera in marine carnivorous fish [53,61], was especially pronounced in the 6% frass group, followed by the 18% and 12% frass inclusion groups (Figure 3c). Some *Vibrio* species have been reported to exhibit probiotic features by assisting in the degradation of dietary materials in the fish gut [61], while other species are known as opportunistic fish pathogens [62,63]. Although the fiber contents were altered among the different diets, there was not a large difference among the bacterial species known to metabolize fiber. The 12% frass inclusion group harbored the most distinct microbiome, characterized by the enrichment of *Cutibacterium, Streptococcus,* and *Bacteroidales p-251-o5* (Figure 3c). *Cutibacterium* members were found to be dominant in the gut of *Dicentrarchus labrax* and *Sparus aurata* individuals fed with insect meal, one of which was derived from *Hermetia illucens* [64], thereby explaining the similarity found in the 12% frass diet. However, the majority of *Cutibacterium* species were accounted for by one of the 12% frass diet replicates (Figure 3a), and may not be an accurate representation of the bacterial composition in all 12% frass diet guts. For the 18S rRNA, 11,017,245 sequences were obtained, which corresponded to 3 ASVs, all of which were annotated as the *Teleostei* genus. A BLAST alignment of these ASVs returned *Trachinotus blochii*, the host fish species used for this study, with 18S ribosomal RNA as the best hit, indicating high sequence recovery of the host DNA (data not shown).

Overall, the 16S pompano gut microbiomes were variable among the different diets, with the control diet showing the largest separation of the community composition. The variability among the frass diets may reflect the gut microbiome of the carnivorous pompano adapting to higher starch- and fiber-containing diets that they are not used to consuming, and may indicate that a longer duration of each feed would be needed to truly see stable differences in the microbiome structure. Future frass feed experiments should focus on increasing the gut microbiome replicates for sequencing and to add metagenomic sequencing to interpret the genomic capabilities of the bacteria present in the guts.

## 4. Conclusions

The present study revealed that dietary frass does not have the potential as a feed ingredient to replace carbohydrate sources in pompano feed. The reduction in fish growth performance with increasing levels of frass was due to the composition of the frass, which appears to not be fully utilized by pompano. It appears that the restricted digestibility or availability of minerals resulted in lower mineral retention in mineral-rich groups (frass diets). Although the pompano gut microbiomes were not vastly different between diets, the high variability indicated potential dysregulation of the gut microflora and may also have led to reduced fish growth performance, although more work would need to be done here to conclude further. However, dietary frass will be considered as a sustainable feed ingredient due to predicted growth in the insect-rearing industry in coming years. Additionally, in this study, the frass was provided from hemp-fed insects, which was not sufficient to maintain growth in a carnivorous species, such as pompano. However, it may still be useful as a sustainable feed ingredient for herbivore and detritivore aquaculture species. Upgrading frass through the modification of the BSF larvae feeding substrate may also be an alternative way to improve the nutritional value and to be able to use frass in combination with high-quality feed ingredients for carnivorous fish diets in the future, which remains to be tested.

## Figures and Tables

**Figure 1 animals-12-02407-f001:**
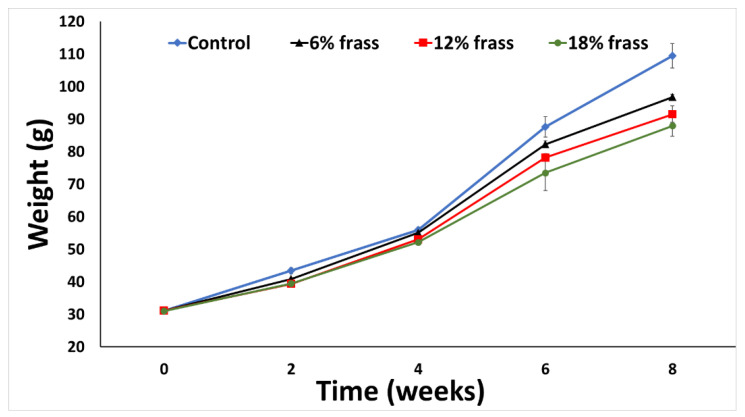
Pompano weight feeding with different levels of frass over the 8 week experimental period.

**Figure 2 animals-12-02407-f002:**
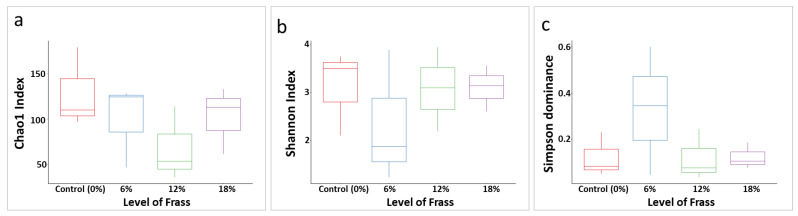
Pairwise ANOVA comparison of the gut biome in fish fed with different levels of frass. (**a**) Chao1 index; (**b**) Shannon index; (**c**) Simpson Dominance.

**Figure 3 animals-12-02407-f003:**
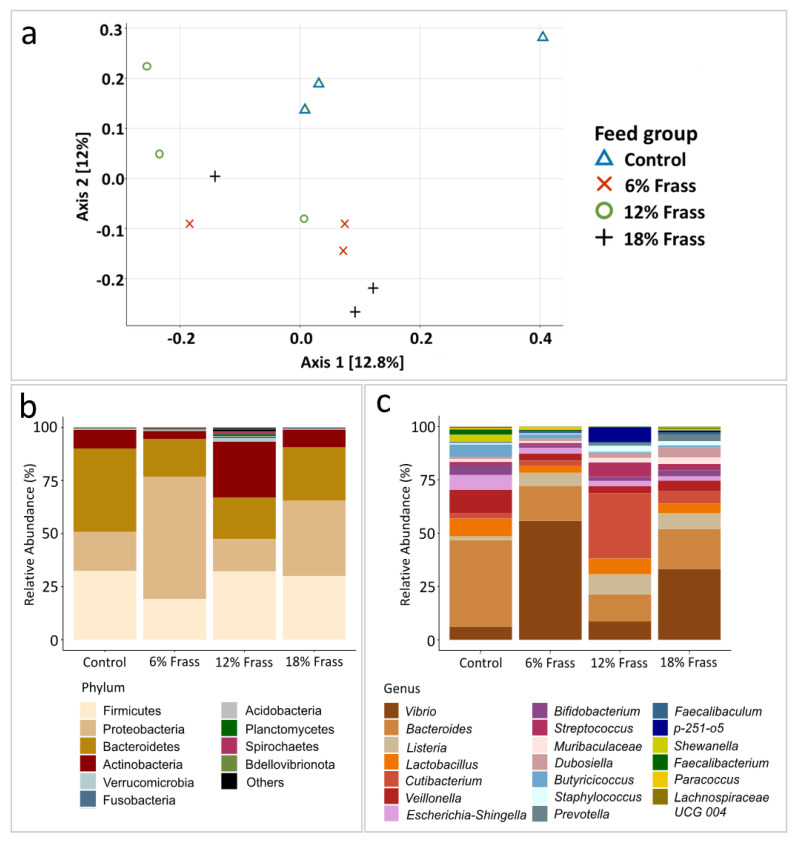
The Florida pompano gut microbial analysis results after feeding with different levels of frass: (**a**) PCA analysis; (**b**) phylum level; (**c**) genus level.

**Table 1 animals-12-02407-t001:** Formulation (g/kg) and chemical analysis of the experimental diets.

Experimental Diets	Control	6% Frass	12% Frass	18% Frass
Ingredients				
Ground corn	104.5	84.5	64.5	44.5
Wheat gluten	80	80	80	80
Frass	0	60	120	180
Fishmeal	200	200	200	200
Wheat flour	190	170	150	130
Soybean meal	110	90	70	50
Fish oil	60	60	60	60
Corn protein concentrate	100	100	100	100
Soy protein concentrate	130	130	130	130
Di. Cal. Phosphate	10	10	10	10
Vitamin C	2	2	2	2
Choline Chloride	6	6	6	6
Vitamin Mix	5	5	5	5
Mineral mix	2.5	2.5	2.5	2.5
Dry matter	918.0	916.3	916.0	912.1
Crude protein	439.7	440.1	427.8	423.0
Crude fat	93.0	91.1	92.0	88.2
Crude fiber	16.1	27.0	38.0	50.6
Starch	227.2	193.7	169.5	135.4
Crude ash	75.8	83.2	88.2	102.9

The vitamin premix consisted of (g/kg premix): thiamin HCL, 0.5; riboflavin, 8.0; pyridoxine HCl, 5.0; Ca-pantothenate, 20.0; niacin, 40.0; biotin, 0.040; folic acid, 1.80; cyanocobalamin, 0.002; vitamin A acetate (500,000 IU g^−1^), 2.40; vitamin D3 (400,000 IU g^−1^), 0.50; DL-α-tocopherol acetate, 80.0; and α cellulose, 834.258. Mineral premix consisted of (g/kg premix): cobalt chloride, 0.04; cupric sulfate pentahydrate, 2.50, ferrous sulfate heptahydrate, 40.0, manganous sulfate anhydrous, 6.50; potassium iodide, 0.67; sodium selenite, 0.10; zinc sulfate heptahydrate, 131.93; α cellulose, 818.26.

**Table 2 animals-12-02407-t002:** Proximate composition (g/kg), amino acid composition (g/kg), and mineral content (mg/kg) of experimental frass.

Nutrient and Mineral Composition	
Dry matter	843.0
Crude protein	106.2
Crude fat	8.0
Crude fiber	236.2
Starch	3.8
Crude ash	149.8
Calcium	19,163.4
Phosphorus	7058.1
Magnesium	4614.7
Potassium	14,786.6
Sodium	7299.9
Iron	1111.9
Manganese	163.9
Zinc	187.5
Copper	26.1
Alanin	8.1
Arginine	4.3
Lysine	5.7
Phenylalanine	2.6
Tyrosine	1.8
Isoleucine	2.1
Leucine	3.9
Cysteine	1.1
Histidine	3.4

**Table 3 animals-12-02407-t003:** The growth performance in Florida pompano juveniles feeding on different levels of frass over an 8 week experimental period. The values are means of triplicate groups ± SD. Means with different letters are significantly different (*p* < 0.05).

Growth Parameters	Diets			
	Control	6% Frass	12% Frass	18% Frass
Initial weight (g)	31.13 ± 0.23	31.2 ± 0.52	31.16 ± 0.49	31.06± 0.41
Final weight (g)	109.5 ± 3.8 ^c^	96.8 ± 0.87 ^b^	91.5 ± 2.53 ^b^	88.1 ± 3.25 ^a^
SGR (%/day)	2.33 ± 0.05 ^c^	2.10 ± 0.02 ^b^	2.0 ± 0.07 ^c^	1.93 ± 0.09 ^a^
FCR	1.28 ± 0.02 ^a^	1.4 ± 0.01 ^c^	1.46 ± 0.06 ^b^	1.5 ± 0.05 ^c^
PER	1.78 ± 0.03 ^c^	1.63 ± 0.02 ^b^	1.61 ± 0.06 ^b^	1.55 ± 0.12 ^a^
HSI%	1.28 ± 0.25 ^b^	0.84 ± 0.12 ^a^	0.95 ± 0.06 ^a^	0.82 ± 0.08 ^a^
VSI%	3.87 ± 0.33 ^a^	4.09 ± 0.84 ^a^	4.47 ± 0.24 ^b^	4.58 ± 0.52 ^b^

**Table 4 animals-12-02407-t004:** Mineral contents of the experimental diets containing different levels of frass.

Minerals (mg/kg)	Diets			
	Control	6% Frass	12% Frass	18% Frass
Calcium	14,963.4	18,552.1	19,152.6	20,556.3
Phosphorus	10,557.1	12,354.8	12,252.1	12,556.3
Magnesium	1514.7	1924.6	2126.1	2416.3
Potassium	7986.6	9436.3	9552.6	10,152.5
Sodium	2799.9	3421.2	3725.1	4256.2
Iron	223.9	292.5	346.5	436.3
Manganese	35.3	43.5	54.1	61.1
Zinc	117.5	133	133.5	132.2
Copper	11.0	12.5	11.5	9.6

**Table 5 animals-12-02407-t005:** Whole-body composition (% of wet weight) and mineral contents (mg/kg wet weight) of Florida pompano fed diets containing different levels of frass.

Experimental Diets	Control	6% Frass	12% Frass	18% Frass
Parameters				
Moisture	68.4 ± 0.7	67.9 ± 0.9	68.3 ± 1.4	68.9 ± 0.7
Crude protein	18.3 ± 0.4	18.6 ± 0.2	19.0 ± 1.2	18.4 ± 0.1
Crude fat	9.8 ± 0.7	9.9 ± 1.0	9.0 ± 0.4	9.0 ± 0.6
Ash	3.4 ± 0.2	3.5 ± 0.1	3.6 ± 0.4	3.4 ± 0.1
Macro mineral content of Florida pompano in mg/g				
Calcium	8.1 ± 0.4	8.5 ± 0.1	8.9 ± 1.1	8.4 ± 0.3
Phosphorus	5.6 ± 0.3	6.0 ± 0.05	6.1 ± 0.3	5.7 ± 0.2
Magnesium	0.4 ± 0.002	0.4 ± 0.01	0.42 ± 0.03	0.41 ± 0.01
Potassium	4.0 ± 0.1	4.1 ± 0.03	4.2 ± 0.3	4.1 ± 0.04
Sodium	1.5 ± 0.1	1.6 ± 0.02	1.5 ± 0.1	1.5 ± 0.02
Trace mineral content of Florida pompano in mg/kg				
Iron	23.7 ± 1.3	29.3 ± 9.8	23.7 ± 2.8	23.1 ± 2.2
Manganese	2.7 ± 0.09	2.8 ± 0.7	3.2 ± 0.4	2.9 ± 0.6
Zinc	21.1 ± 0.6	22.43 ± 0.4	22.0 ± 1.4	21.4 ± 1.4
Copper	1.0 ± 0.1	2.4 ± 2.4	1.36 ± 0.4	1.4 ± 0.3

## Data Availability

Data will be available on demand and will be shared with the public.

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
