# Peer review of "Nutritional Evaluation of Black Soldier Fly Frass as an Ingredient in Florida Pompano (*Trachinotus carolinus* L.) Diets"

_animals, 2022, doi:10.3390/ani12182407_

Round 1

Reviewer 1 Report

Dear authors,

this article is interesting in its experimental procedure, but it contains some fundamental aspects to be clarified before the possible publication.

The BSF frass proposed here has a small amount of protein, but you need to add the amino acid composition of this protein. The protein content only is not sufficient to justify the inclusion, considering the variability of this feedstuff that you correctly mentioned.

BSF amino acid composition table

Considering that the Florida pompano is a carnivorous species, the amount of crude fiber in the propose fish feeds is impressive, in particular in the highest level of inclusion (50%). You compared your results with those of tilapia and catfish and you correctly explained that these latter species are omnivorous.

Is there any reference (both scientific or commercial) about a suitable fish feed for Florida pompano with comparable levels of crude fiber, ranging from 16 to 51% ?

Why did you used this level so high of crude fiber, knowing that for a carnivorous fish it could be easily refused, independently from protein source used? You need to add scientific references that could support your decision to use these fish feed formulations.

9% of crude fat is a reasonable level for Florida pompano or for another carangid? Please, include a reference on this aspect. Otherwise 15% of crude fat is the average level for a pelleted feed for carnivorous species.

Authors should explain the difference in the interpretation of diversity indexes used in the study of gut biome, in particular the Chao1 index. What is the different meaning of those indexes?

I think that this paper could be published only after a clarification of these aspects and some extra analyses on BSF frass amino acid composition.

Author Response

Thank you, we revised the article based on your comments. 

Reviewer 2 Report

Dear editor

Pls see the attached file.

REGARDS

Author Response

Thank you, we revised accordingly. 

Reviewer 3 Report

General Comments: the article investigates the potential of insect fras (from BSF) as a feed ingriedent for carnivorous fish, which is a innovative and original area of research. The justification for the research, i.e. using frass as a feed ingriedent, is somewhat confused. The problems associated with fish meal are stated; yet, then frass is listed as a carbohydrate feed ingriedent (which is correct). Please rephrase the argumentation/justification of this research to match the characteristics of frass. It’s inclusion in fish diets is not likely to minimize the need for fish meal; this is apparent within the experimental diets. The text, although the language used is mostly clear throughout, should be proofread, preferrably by an english native speaker, e.g. there are missing noun articles and pluralizations (or pluralizations where not needed). I have pointed out only a handful of the mistakes within the manuscript (see specific comments). There are major concerns regarding the experimental design and how  it was conducted, as well as how data were analysed. First, the text describing the experimental design is unclear; leading to the conducted experiment being unclear. Second, the adaptive feeding based on animal growth could bias the results of this experiment, as the frass-fed animals grow more slowly they are given less feed in the coming weeks of the experiment, which may compound the slow growth effect (due to diet). This should be thoroughly discussed in the discussion, as well as the choice of adaptive feeding being reasoned in the Materials and Methods section. Third, the statistical analysis applied are not sufficient regarding the described experimental design. Fourth, sample sizes per parameter should be listed directly throught the manuscript, preferrably in the tables in the results, i.e. n=x for the parameter final weight within each treatment -> I believe this is n=30. Doing so increases transcparency and understanding.  

Specific comments: 

Simple summary, Abstract, Introduction: please delete the content refering for the need for protein feed ingriedents, such as fish meal. Frass is not a high protein ingredient. As exemplified in the experiment, frass is likely only to substitute plant carbohydrate ingredients in carnivorous fish diets. (E.g. Lines 21-23, 35-36, 65-66)

Line 55: “…a supply…”

Line 56: “Partial of complete substitution…”

Lines 65-66: It is true that the rearing of insects for aquaculture production and then including the byproduct frass in the diet (with the insect meal as protein source) could be an interesting and sustainable system. However, this is not the experiment carried out here. I suggest removing the sentence entirely, as insect meal is entirely different from insect frass.

Line 76: the taxanomic species name should be in italics

Line 77: “rapid growth with commercial diets, relatively high prices, and…”

Line 83: “…larval frass from BSF production has been shown to have a positive significant effect…”

Line 94: please use institutions full name. 

Line 97: please include company and version information for Lindo software

Line 97: please include the modelling parameters used to calculate diets. To my knowledge, Lindo is not a rations calculator program, so information regarding balancing parameters (e.g. crude protein, energy, etc.) is needed along with statistical function information. 

Line 107: This sentence is phrased incorrectly. It says that only a total of ten fish were used in your experiment. I think what was meant was: 120 fish were allocated to 12 tanks (10 fish pro tank; average body weight XY pro tank). Three tanks were designated per treatment group. 

Line 114: remove “each in a completely randomized tank design”. This is not an experimental design construct. Unless statistical software was used to generate the allocation of which tank was assigned to which group, it is not a random design. 

Line 115: The adaptive feeding based on animal growth could bias the results of this experiment, as the frass-fed animals grow more slowly they are given less feed in the coming weeks of the experiment, which may compound the slow growth effect (due to diet). This should be thoroughly discussed in the discussion, as well as the choice of adaptive feeding being reasoned in the Materials and Methods section.

Line 117: some word/something is missing in this sentence. It is incomplete.

Table 1: if the frass was meant to substitute carbohydrate ingredients, what was done to try and balance overall energy in the diets? Please include energy in Table 1, or at the very least an explanation in the text, i.e. was the aim to have iso-energetic diets?

Table 1: please include if the g/kg is referring to dry or fresh matter

Table 2: as dry matter is listed, what is the g/kg referring to? Is it g/kg fresh matter? Please include this in the table description. 

Line 154: I assume the reference to AOAC needs to be removed. If the procedure outlined by AOAC was not used, then it should not be mentioned. It is perfectly fine to just state how things were conducted (without a reference), if the procedure is not a standardized one. 

Line 156: I think with “sub-samples” you are referring to sample replicates (per animal). 

Table 3: the contents of Table 3 should be inserted into Table 1.

Section 2.7: as there are at least three (animal) replicates per tank, a possible tank effect should be incorporated into the statistical analysis. Please re-do the analysis as a two-way ANOVA where the tank in which an animal was reared is included as a fixed effect. Using such a model may allow for a more complete interpretation of the data, such as an explanation of high heterogeneity within a single treatment group.   

Line 267: unless a statistical correlation was conducted, please remove the word “correlation” and use “followed” or “trended with”, etc. If a correlation was conducted, please include the p-value/significance level. 

Line 309: “those of” -> as you cite two articles, it implies that the findings are plural. 

Line 392: please replace “will” with “can”

Line 393 - 396: perhaps state this more generally - Insect rearing-substrate effect frass quality and therefore more research is needed to ensure high quality frass. 

Line 396: replace “upgrading” with “improving frass quality”

Line 397: “substrates”

Line 398: “be able to use frass as a high-quality…”

Round 2

Reviewer 1 Report

Dear Authors,

following my previous suggestions, you added clarifications that make this paper worth of publication.

Author Response

Thank you

Reviewer 3 Report

Dear Authors, 

thank you for making the suggested changes. I wish you all the best!